# “Maternal Vaccination Greatly Depends on Your Trust in the Healthcare System”: A Qualitative Study on the Acceptability of Maternal Vaccines among Pregnant Women and Healthcare Workers in Barcelona, Spain

**DOI:** 10.3390/vaccines10122015

**Published:** 2022-11-25

**Authors:** Anna Marín-Cos, Elena Marbán-Castro, Ivana Nedic, Mara Ferrari, Esther Crespo-Mirasol, Laia Ferrer Ventura, Berta Noya Zamora, Victoria Fumadó, Clara Menéndez, Cristina Martínez Bueno, Anna Llupià, Marta López, Anna Goncé, Azucena Bardají

**Affiliations:** 1ISGlobal, Hospital Clínic-Universitat de Barcelona, 08036 Barcelona, Spain; 2BCNatal—Barcelona Center of Maternal-Fetal and Neonatal Medicine, Department of Maternal-Fetal Medicine, Hospital Clínic and Hospital Sant Joan de Déu, Universitat de Barcelona, 08007 Barcelona, Spain; 3ASSIR Esquerra, Gerència Territorial de Barcelona, Institut Català de la Salut, 08015 Barcelona, Spain; 4Centro de Investigação em Saúde de Manhiça (CISM), Maputo 1929, Mozambique; 5Consorcio de Investigación Biomédica en Red de Epidemiología y Salud Pública (CIBERESP), 28029 Barcelona, Spain; 6ASSIR Barcelona Ciutat, Gerència Territorial de Barcelona, Institut Català de la Salut i Universitat de Barcelona, 08007 Barcelona, Spain; 7Department of Preventive Medicine and Epidemiology, Hospital Clinic, Universitat de Barcelona, 08036 Barcelona, Spain

**Keywords:** maternal vaccines, acceptability, pregnancy, hesitancy, barriers, facilitators, clinical trials

## Abstract

The World Health Organization (WHO) identified vaccine hesitancy as one of the top 10 threats to global health in 2019. Health promotion and education have been seen to improve knowledge and uptake of vaccinations in pregnancy. This qualitative study was conducted based on phenomenology, a methodological approach to understand first-hand experiences, and grounded theory, an inductive approach to analyse data, where theoretical generalisations emerge. Data were collected through semi-structured interviews with pregnant women attending antenatal care services and healthcare workers (HCWs) in Barcelona, Spain. Interviews were audio-recorded, transcribed, and coded, and notes were taken. Inductive thematic analysis was performed, and data were manually coded. Pertussis was reported as the most trusted vaccine among pregnant women due to its long-standing background as a recommended vaccine in pregnancy. The influenza vaccine was regarded as less important since it was perceived to cause mild disease. The COVID-19 vaccine was the least trustworthy for pregnant women due to uncertainties about effectiveness, health effects in the mid- and long-term, the fast development of the vaccine mRNA technology, and the perceptions of limited data on vaccine safety. However, the necessity to be vaccinated was justified by pregnant women due to the exceptional circumstances of the COVID-19 pandemic. The recommendations provided by HCW and the established relationship between the HCW, particularly midwives, and pregnant women were the main factors affecting decision-making. The role of mass media was perceived as key to helping provide reliable messages about the need for vaccines during pregnancy. Overall, vaccines administered during pregnancy were perceived as great tools associated with better health and improved quality of life. Pregnancy was envisioned as a vulnerable period in women’s lives that required risk–benefits assessments for decision-making about maternal vaccinations. A holistic approach involving the community and society was considered crucial for health education regarding maternal vaccines in support of the work conducted by HCWs.

## 1. Introduction

The World Health Organization (WHO) identified vaccine hesitancy as one of the top 10 threats to global health in 2019 [1]. The success of vaccines depends not only on their demonstrated efficacy or cost-effectiveness, but also on their acceptability by the general population [1]. Individuals may accept some vaccines but reject others, while personal beliefs may change over time. Hence, vaccine hesitancy is not always a total refusal but is commonly seen as a gradient of responses [2]. Evidence is discordant on whether a population’s education is associated with higher or lower vaccine hesitancy in pregnancy. A study conducted in Europe concluded that health promotion and education increased knowledge about influenza vaccination in pregnancy [2]. Another study in the UK suggested that disease risk information did not influence vaccine acceptability in their sample of pregnant women [3]. There are knowledge gaps in educational levels and health literacy concerning vaccination.

Since 2005, under the guidance of the Global Advisory Committee on Vaccine Safety, WHO has recommended the administration of the inactivated flu vaccine and the Tdap (tetanus, diphtheria, and acellular pertussis) vaccine in pregnant women [4]. Pregnant women are at a greater risk of harmful pregnancy outcomes when contracting influenza during gestation, with increased severity in later stages of gestation, and their infants of a higher risk of intrauterine foetal deaths, preterm births, and low birth weight [5,6]. The currently recommended maternal trivalent inactivated influenza vaccination was first recommended in 2012 [5], and extensive research found no safety concerns [7]. In Spain, the newest data on influenza vaccine uptake in pregnant women from the Ministry of Health show 61.9% coverage for the 2020–2021 season, varying from 17.3% in Ceuta to 91.1% in Valencia [8].

Maternal Tdap vaccination is recommended after 27 weeks of gestation to decrease the incidence of pertussis infection in infants under six months of age [9]. Pertussis, also known as whooping cough, is among the most prevalent vaccine-preventable diseases (VPD) in Western countries, with multiple epidemics documented in the past decade [10]. Five European countries, including Spain, concentrate 72% of pertussis cases in Europe, with the number of reported cases, though underestimated, increasing annually [11,12]. Maternal Tdap vaccination rates show significant variability between and within countries, with coverage in Spain reaching 89.8% in 2021 [8]. In the United States and Europe, stark contrasts were noted in the acceptance of the maternal COVID-19 vaccination, with coverage rates between 29.7% and 77.4% correlated with maternal influenza vaccine uptake [13]. While both influenza and pertussis vaccinations are considered safe during pregnancy by healthcare workers, there is a disparity in knowledge about these vaccines, leading to a reduction in healthcare worker recommendations [14].

Furthermore, in light of the SARS-CoV-2 pandemic, in January 2021, WHO developed interim recommendations on COVID-19 vaccination for pregnant and lactating women under emergency use authorisation based on risk–benefit analyses [15]. Despite differences across European countries, COVID-19 vaccine hesitancy was identified among almost half of the 16,000 pregnant and lactating women surveyed in Belgium, Norway, the Netherlands, Switzerland, Ireland, and the UK [16]. During the COVID-19 pandemic, researchers advocated for including pregnant women in COVID-19 vaccine trials [17,18]. However, despite pregnant women being at increased risk of adverse outcomes from COVID-19, they were excluded from COVID-19 vaccine trials. During the pandemic, studies have documented low acceptability among pregnant women towards receiving COVID-19 vaccines [19] or participating in drug-based clinical trials to prevent COVID-19 [20]. The paucity of data demonstrating COVID-19 vaccine safety, immunogenicity, and efficacy in pregnant and breastfeeding women made women’s decisions to get vaccinated a significant challenge [21].

This article aims to understand further pregnant women’s and healthcare workers’ (HCWs) perceptions about maternal vaccines, the primary motivational factors for decision-making regarding the uptake of maternal vaccines, and the participation in clinical trials during pregnancy.

## 2. Materials and Methods

### 2.1. Study Design and Population

This qualitative study was conducted based on phenomenology (methodological approach to understand first-hand experiences) [22] and grounded theory (inductive approach to analysing data, where theoretical generalisations emerge) [23]. The study was conducted in Spain between January and June 2022 among populations living in the Barcelona metropolitan area. The study participants were pregnant women, and HCW providing clinical and/or preventive services to pregnant women. The inclusion criteria for pregnant women were being pregnant, of any gestational age, and being willing to be interviewed. Inclusion criteria for HCWs included being an HCW working at the Maternal Foetal Medicine Department of the Hospital Clínic of Barcelona (HCB) or the Primary Healthcare Centres providing antenatal care and reproductive health services in the Barcelona metropolitan area (Atenció a la Salut Sexual i Reproductiva-ASSIR). Study participants were recruited through convenience sampling. The sample size was defined based on a saturation point whereby all themes had been thoroughly explored, and no new themes emerged in subsequent interviews. During the coding process, and while comparing categories that emerged during the interviews, as all themes emerged in both groups, the saturation point was achieved; no more interviews were scheduled to be performed [22,23,24].

### 2.2. Data Collection

Data were collected through semi-structured interviews and field notes. The design of the interview guide for pregnant women (See Appendix A) was based on the following topics: (1) knowledge about maternal immunisation, (2) experiences of receiving vaccines during pregnancy, (3) experiences with the COVID-19 vaccine, and (4) women’s hypothetical participation in clinical trials (see Appendix A). For HCWs, the interview guide (See Appendix A) included topics related to their practice regarding providing information about vaccines administered during pregnancy. Interviews were performed in Spanish or Catalan and lasted approximately 20–40 min each. A baseline interview with demographic data was collected prior to the interview. One experienced anthropologist performed the interviews. All interviews were conducted remotely through video calls through online platforms or phone calls, according to participants’ preferences. All interviews were audio-recorded, and notes were taken. Transcriptions were performed with the assistance of an online transcriber (Sonix, 2022, San Francisco, CA, USA). Researchers regularly discussed key findings, difficulties, and any changes needed to the data collection guides according to emerging data.

### 2.3. Data Analysis

Audio recordings were summarised, and particular pieces related to the topics of interest were transcribed. Inductive thematic analysis was performed, and data were manually coded. An initial coding frame was developed based on themes arising from the analyses of the first transcripts using open and inductive coding. Then, a consensus on codes and emerging themes was reached between two investigators (A.M. and E.M.-C.). A final coding frame was agreed upon, and themes and categories were grouped. This article has been prepared per reporting standards in the SRQR guidelines for reporting qualitative studies [25].

### 2.4. Ethical Considerations

Ethical approval for this study was granted by the Ethics Review Committees of the Hospital Clínic Barcelona (CEIm) (Reg. No. HCB/2021/0352) and by the Primary Health Care Network Ethics Committee (CDEI) IDIAP Jordi Gol (Reg. No. CEI 21/149-P). The study was conducted under the Good Clinical Practice Guidelines, the Declaration of Helsinki, and local rules and regulations. Participants gave oral consent for interviews and audio recordings. All names and other personal identifiers in the transcripts were deleted to guarantee subject anonymity. For this purpose, codes were generated. Interviewers’ codes ending in “01” were given to pregnant women, and those ending in “02” were for HCWs.

## 3. Results

### 3.1. Participants’ Characteristics

A total of 35 participants were included in the study, 21 pregnant women and 14 healthcare professionals. Their sociodemographic characteristics are presented in Table 1. For pregnant women participants, most (61.9%) were Spanish, with a mean age of 33.7 years, and most of them were employed (90.5%) and held a university or postgraduate degree (61.9%). Regarding their marital status, the majority of women were in a relationship (42.9%), in a domestic partnership (14.3%), or married (28.6%). Most women had a gestational age of 24 weeks or more (61.9%) at the time of the interview, and the majority (71.4%) had been pregnant before. Ten out of the twenty-one women reported having experienced COVID-19 disease before their current pregnancy, while ten women had COVID-19 during their current pregnancy. As for HCWs, the majority (42.9%) were 46 years old or older, and most were Spanish (85.7%). Half of the HCWs interviewed were midwives (50%), followed by obstetricians and gynaecologists (35.7%).

### 3.2. General Perceptions of Vaccines

Pregnant women perceived vaccines as necessary for adults and children, but there were some concerns regarding the risks during pregnancy. Some women explained that vaccine recommendations during pregnancy should be adapted depending on each case after a risk–benefit assessment. HCWs also highlighted the importance of adapting vaccine administration to the population and epidemiological context. They pointed out that the adverse effects that pregnant women have are not much different from non-pregnant people. HCWs felt comfortable administering vaccines during pregnancy, except for the COVID-19 vaccine, about which they were more doubtful. Pregnancy was seen as a vulnerable period by both groups interviewed. Therefore, greater confidence and trustworthiness were needed when deciding to get vaccinated.

Additionally, many women justified their decision to get vaccinated precisely because they felt more vulnerable to diseases. Women explained that they were cautious about taking drugs during pregnancy. For that reason, some women were also careful and tried to avoid vaccines, while others preferred to get vaccinated, as they would have to avoid drugs if they became infected.


*“I’m always afraid that there is some risk, to be honest (…) I know that during pregnancy, you can’t even take ibuprofen (…) So, it seems extreme to me that suddenly they vaccinate you, they put many more things in your body than they usually advise. (…) I said yes when they explained it to me, as I see that it is something that has been experimented with for a long time and many pregnant women do it”.*
105-01 (Pregnant woman)

Some women believed that if the disease is mild and there are secondary side effects of the vaccine, they would prefer to bear the risks of not getting vaccinated and instead be naturally immunised if they get infected. The perceived trust or distrust of the health system is directly related to the perceived safety of vaccination. Among the HCWs interviewed, a certain distrust of pharmaceutical companies conducting vaccine trials also arose, related to the lack of clarity when reporting side effects, negatively affecting vaccine confidence.

HCW discussed the need to adapt vaccination coverage to each woman’s background and vaccination history, considering their specific risks of developing the disease.

### 3.3. Decision-Making Factors for Vaccination during Pregnancy

When deciding whether to get vaccinated, some women made their decision before attending antenatal care visits and did not consider other opinions. These women, especially those who decided not to get vaccinated, emphasised that it was their own personal decision. Most pregnant women stated that they trust and follow the HCWs’ recommendations and would probably not consult other sources of information. Other pregnant women needed to dwell on the decision or consult a family member and even seek more information from other sources beyond relying on medical recommendations.

Table 2 and Table 3 present respectively, motivational factors for deciding to be vaccinated during pregnancy or not to be vaccinated. See Appendix A with the original quotes from Table 2 and Table 3.

#### 3.3.1. Factors Influencing Decision-Making Regarding Maternal Vaccinations

Pregnant women’s trust in HCWs and medical authorities is essential in their decision-making process to be vaccinated. Women stated that having all the information about a specific vaccine and the advantages and disadvantages of vaccination was important for them. The fact that the risks were mentioned together with the benefits transmitted certain confidence. Parallel to this, and as a source of distrust, there were non-experts (people outside the field of healthcare) who generated debate about vaccinations in the media. A common point on which the HCWs and women interviewed coincided was the influence of the personal opinions of the HCWs on women´s decisions about whether to get vaccinated or not. The information provided during the antenatal care visit and the relationship between the HCW and the pregnant women were also important factors (see Appendix A).

#### 3.3.2. Healthcare Professionals’ Perceptions of Pregnant Women’s Decisions to Be Vaccinated

When HCWs were asked whether they believed that there was a profile of women more oriented towards vaccination than others, several profiles of women emerged. Within these profiles, the following influential socio-cultural factors came to light (see Figure 1).

Among the women who decided to get vaccinated, there was a profile of women who did not question the vaccination and followed HCW’s recommendations. These women would not consult more sources of information either. Within this group, oriented towards accepting vaccinations, there was another profile of women with higher education in the scientific or medical field who favour vaccination and, thus, would look for more information. Among the women who had doubts regarding vaccination in pregnancy and women who came to the consultation with doubts, the opinion of the HCW, their relationship with them, and the information presented in the consultation room would play a key role.

Among the women who came to the consultation without having made a clear decision, there was a profile of women with higher education who were not involved in the health or medical field who would come to the consultation with information that they had acquired themselves (not always truthful), and who would also ask for more information. This group of women would decide based on the arguments presented by the HCW in question and on the information obtained outside the consultation. Among the women oriented towards not being vaccinated, there was a specific profile of native Spanish women with higher education, a medium-high socioeconomic level, and an inclination towards what they consider “a more natural pregnancy”. This profile of women may have a lower perception of risk for infectious diseases. Among the women in this group oriented towards not being vaccinated, there were also non-native Spanish women with a significant cultural or language barrier. The HCWs also agreed that in the case of women who arrived at the consultation having already made the decision not to be vaccinated, it was almost impossible to change their minds (see Figure 2).

Although most HCW agreed that non-native Spanish women are generally more likely to be vaccinated, a small percentage of professionals commented that a language barrier linked to a cultural distance could lead to women declining vaccines. The same can be observed with the level of education. Women with a higher education sometimes appear likely to be vaccinated due to a supposed “understanding” of the need for vaccination, while other times, this appears as a source of doubt that moves them away from deciding in favour of vaccination.

#### 3.3.3. Vaccines: Opinions, Credibility and Trust

Pertussis was the most trusted vaccine by pregnant women and HCW due to the long-term use of this vaccine during pregnancy and its efficacy in preventing the risks of the disease in infants. One uncertainty that emerged from the interviews was that the pertussis vaccine is not manufactured alone but together with diphtheria and tetanus (see Appendix A). Some pregnant women and HCW argue that certain doubts emerge when the women are informed that the pertussis vaccine is administered with tetanus and diphtheria.

Influenza vaccination was generally perceived as less necessary, as it is not considered a severe disease, and women preferred to bear the risks of the disease rather than the side effects associated with the vaccine. The influenza vaccine was also perceived as less effective because it does not prevent infection and because the virus mutates yearly, and the vaccine needs to be changed (see Appendix A).

The COVID-19 vaccine was perceived as generating the most mistrust because it was less tested, and the long-term side effects are still unknown. It was also argued that it had been produced very quickly and was, therefore, something new. It was reasoned that it is a vaccine generated from a different technology than the other two vaccines administered during pregnancy. Some women felt like subjects of experimentation. Parallel to these explanations, the COVID-19 vaccine was perceived as having low efficacy because people got infected anyway. When deciding to be vaccinated against COVID-19, one reason was “to do good for society”. Other women questioned this position and showed indignation at this idea, encouraging vaccination as an altruistic gesture. In general, the main reason for not getting vaccinated was fear and distrust of the vaccine and the health and/or political system, which in this case go hand in hand since the health measures have been made with politics in mind. Some women criticised the idea of “generalised vaccination” as opposed to testing antibodies or looking at the suitability for administering vaccines in pregnancy case by case before vaccinating to detect the real need according to each individual. To this is added the disorientation and uncertainty caused by changes in vaccination guidelines and the reality of the lack of unanimity in the recommendation of this vaccine. Another issue that stands out at this point is the perception of complications during pregnancy and stories of miscarriages coinciding with COVID-19 vaccination during pregnancy. All these reasons were understood and primarily justified by the fact that this was an exceptional situation: a pandemic where actions were taken based on the immediate and changing reality.

### 3.4. COVID-19 Vaccination in Pregnancy

#### 3.4.1. Recommendation of the COVID-19 Vaccine

In general, HCWs felt they lacked sufficient scientific information to recommend the COVID-19 vaccine in pregnancy and felt insecurity, uncertainties, and even anxiety about recommending it. As the pandemic evolved, the confidence to recommend it increased for certain HCWs. Others believed that even today, it is still impossible to know the side effects, as not enough time has passed. Secondly, the lack of COVID-19 vaccine clinical trials on pregnant women generated mistrust (see Appendix A). Thirdly, they claimed there had been inaccurate reporting of side effects due to an inefficient system. Faced with this lack of information, many HCWs followed the guidelines with doubts. Most agreed that the guidelines were brief, and some viewed them as ambiguous. Other HCW avoided recommending the vaccine, as they expressed having a moral conflict. HCWs generally recommended other preventive measures to diminish COVID-19 exposure and potentially avoid infection. When professionals talked about COVID-19 vaccination recommendations, there seemed to be an official recommendation (guidelines) and a more subjective one that appeared to be more critical with the vaccination depending on the trimester of pregnancy (see Appendix A).

Currently, the administration of this vaccine is recommended at any stage of pregnancy. Despite this, most of the professionals interviewed disagreed with this recommendation, and “as a precaution”, they preferred to avoid recommending the vaccine in the first and third trimesters. This was partially caused by all the changes in the guidelines, especially at the beginning of the pandemic. In some cases, HCWs have placed the ethical responsibility on the institutions in charge of developing and supporting protocols and guidelines. HCWs stated that they attended meetings where the guidelines were discussed. Some also searched for scientific evidence to become more informed on where the protocols come from. All the professionals agreed that the guidelines must be followed and that everyone gives or should give the same information.

In contrast, there was the idea that “everyone knows what happens in the consultation room”. Some HCWs reported on it objectively, and others stated that they have “freedom” and room for manoeuvre. When talking about COVID-19, there was a much greater permissiveness when it came to hindering its recommendation, not mentioning it, or recommending it at more limited times than the protocol proposes. The changes in the COVID-19 vaccination protocols did not help the feeling of the need to follow the guidelines unanimously. These changes also led to doubts and insecurities in pregnant women who were faced with deciding whether to be vaccinated. The changes in the protocol were perceived as a factor that favours the feeling of permissiveness in the recommendation (see Appendix A).

Some women recounted how the health staff recommended waiting to be vaccinated and insisted that the decision was up to them. They felt conflicted as they were receiving different and practically opposite recommendations depending on the institution or professional visited. Some women associated miscarriage as a side effect of the vaccination if they suffered a miscarriage shortly after being vaccinated. According to some interviews, certain women were vaccinated during pregnancy, while others were advised to wait until after the child was born.

#### 3.4.2. Perceptions of the Role of Health Authorities

Some women thought that the role of health authorities should be linked to informing or recommending vaccines objectively. Nevertheless, other women complained that when protocols were followed, recommendations were made in a generalised way instead of involving an individualised approach. There would be more demand for explanations and information from the health sector. Since the onset of the pandemic, different stances on vaccination have become evident. This impacted confidence in the vaccination recommendations. Debate was generated, and people were questioning the guidelines instead of simply following the recommendations as in the past. Some people interpreted this debate about vaccination as being linked to awareness and increased knowledge, both those who choose to get vaccinated and those who did not. Other people had the perception that from now on, vaccine recommendations would be more effective. Others thought there had been an increase in “anti-vaccine” movements, which can be seen as privileged because those populations have access to the healthcare system and infectious diseases are not very prevalent in their societies. COVID-19 vaccination was also a political issue and a question of trust in authorities: people who perceived authority as a risk and who may have felt that their freedoms were being taken away during the pandemic found it easier to oppose certain measures. Conversely, people who considered authorities as guarantors of public health and protective of their communities would more easily follow vaccination guidelines and other recommendations.

#### 3.4.3. Sources of Information and the Influence of the Media

Most pregnant women considered the midwives who monitor their pregnancy as their primary source of information for vaccines. To a lesser extent, some women looked for information from other reliable sources. Some women in the sanitary or research field mentioned consulting academic sources. Women emphasised the importance of differentiating between reliable and unreliable sources. Some women who looked for information on the Internet warned about the danger of believing everything on social media. Both women and HCWs believed that the media greatly influenced the decision to vaccinate, especially concerning COVID-19. Some HCWs complained about it and emphasised the need for more comprehensive health education in society, schools, and the media. Although COVID-19 vaccination was thought to be mediatised and to generate debate, most of those interviewed believed that the role of the media has been in favour of vaccination.

### 3.5. Clinical Trials during Pregnancy and Motives to Participate or Not

Reluctance to participate in a clinical trial was more significant than the willingness to participate. The main reason was uncertainty or lack of security during pregnancy. Among the risk–benefit ratio, there was a much greater perception of risk than benefit. An underlying fear even made interviewed women defensive when answering this question. Some women judged that clinical trials with pregnant women were considered, while at the same time argued that they would not be vaccinated for lack of clinical trial data. Parallel to this feeling of fear, some women empathised with the need for research and clinical trials but recognised much greater personal fears that would prevent them from participating. Some HCWs stated they would not participate if pregnant because they feared for their safety. The HCWs recognised that it is difficult for pregnant women to participate in a clinical trial precisely because they are pregnant. Even so, it is emphasised that there is variability depending on the trial in question and its implications. Conversely, according to the pregnant women interviewed, one of the reasons to participate in a clinical trial would be if HCWs recommended the vaccine or if the reasons explained to them for participating were convincing. On the other hand, a relationship between participation and being able to contribute to scientific knowledge is also evident. The possible benefits of the vaccine in the trial also emerged as a motivational factor. The HCWs participating in clinical trials while pregnant argued that depending on the study, they would think about it, since their experience could be helpful to other women. However, the specific trial appears as a decisive factor in the decision-making process. According to the HCWs interviewed, one reason for the women to participate would be if there were health rewards for the women. Most of the women interviewed found it difficult to imagine situations in which they would participate. When these situations were considered, they recognised they would accept to participate, making it clear that these are exceptional situations (risk to the foetus, the mother, and the pandemic situation…). Among the exceptional situations in which women would accept to participate, there was the hypothetical situation in which the foetus presents some disease that could only be saved with a clinical trial. To participate, the women would need a guarantee that the trial would not affect the foetus negatively (see Appendix A).

## 4. Discussion

The importance of this study is vested in the rich, in-depth information obtained from pregnant women and HCWs around diverse topics related to vaccination. In general, vaccines were perceived as a significant medical advancement associated with increased life expectancy and quality of life. However, pregnancy was conceived as a vulnerable period in women’s lives, where risk–benefit assessments need to be performed to decide whether to be vaccinated or participate in a clinical trial. One of the most important factors for decision-making was HCW opinions, the information given in healthcare centres, and the relationship with the HCW, especially midwives, during antenatal care visits. In line with other authors, the vaccine confidence strategy reinforced confidence in vaccines, including building trust, empowering HCW, and engaging communities and individuals [26]. Our results indicated that some potential profiles of women were more prone to be vaccinated, depending on what information was perceived to be more trustful. Socio-cultural factors might have influenced decision-making, such as country of origin, language, educational level, education field, socioeconomic status, trust in the healthcare system, beliefs, self-perception of risks of acquiring the disease, and development of secondary effects from the vaccine, etc. Attitudes to COVID-19 vaccination during pregnancy may be modifiable by the design of the information provided concerning women’s education, occupation, and parity. Longitudinal studies are needed to determine whether this approach can effectively increase the rates of COVID-19 vaccination in pregnant women [27].

This study was performed during the COVID-19 pandemic, which inherently conditioned interviewers’ responses and general perceptions about vaccines. The inclusion of the COVID-19 vaccine as a topic in the interviews enabled further discussions related to healthcare protocols, participants’ feelings, and other factors for decision-making. It also allowed us to explore the primary sources of information and the role of mass media in COVID-19 vaccination campaigns. Pregnant women were worried about vaccine effects during pregnancy and breastfeeding. Some of them expressed willingness to receive more vaccine safety and efficacy information. A study by Lily Huang et al. illustrates a wide range of perspectives regarding COVID-19 maternal vaccination, with many citing concerns over the consequences on their child or the lack of information in the perinatal period [28]. Perinatal women were also faced with the decision to receive a COVID-19 vaccination without a clinical trial on vaccine safety and efficacy [28]. Similarly, COVID-19 vaccine hesitancy was frequent (46%) in a cross-sectional study performed with pregnant and post-partum women in Ohio (USA) [29] and in a cohort study performed in London (USA) where less than one-third of pregnant women acceded COVID-19 vaccination [30]. In Texas (USA), a cross-sectional survey study on the sociodemographic predictors for COVID-19 vaccine hesitancy in pregnancy found that women who were hesitant were younger, had a more advanced pregnancy, and were also hesitant about flu and Tdap vaccination [31]. Additionally, they reported not having received enough information to make their decision [31]. Interestingly, a survey conducted in Italy showed that the pandemic experience could positively change attitudes toward immunisation in pregnancy, as it raised awareness of the role of vaccines [32].

Our results highlighted the uncertainties and distress felt by HCWs to recommend the COVID-19 vaccine without proper scientific evidence based on clinical trials. Several continuous changes in medical protocols, and ambiguous guidelines, reduced HCW trust in recommendations, leading to feelings of insecurity about providing those recommendations. These changes also increased the freedom with which HCWs recommended vaccination, as guidelines evolved in terms of when to recommend vaccination (weeks of gestation), leading to HCWs providing their personal recommendations and not following the most current guidelines. In line with our results, a study in Italy concluded that vaccination hesitancy could be minimised by consistent recommendations to all pregnant women by obstetric staff during antenatal care visits [32].

Perceived risk of the virus and public trust played critical roles in shaping vaccine acceptance and confidence on top of the pre-COVID-19 vaccine attitudes [33]. The perceived risk of COVID-19 for the population, as measured by their worries, and belief in the importance of having a vaccine and mass vaccination, is a stronger predictor of COVID-19 vaccine acceptance and confidence compared to the perceived risk of infection for themselves or their children [33]. The hesitancy of the disease, scepticism of a new vaccine, and distrust of the system, along with previously shaped vaccination beliefs, contributed to vaccine reluctance [33]. All parameters are dynamic and influenced by current events, however, their roles follow already-known general factors for vaccine hesitancy in pregnancy [33]. It is vital to bear in mind that the issue of COVID-19 vaccination is linked to political and social measures that have been guiding and managing the pandemic. Therefore, the personal perception of individuals about those people or groups they perceive as authorities—such as politicians, the media, and health authorities—may have had a direct influence on opinions regarding vaccination. Vaccination entails a complex issue among the need to protect collective health and the individual right to freedom of choice [34]. In Europe, vaccination is considered an individual choice based on personal conviction, thus ruling out generalised vaccination mandates [34].

In general, vaccines in pregnancy were perceived as necessary after a positive risk–benefit assessment. However, some population sub-groups, especially those women with higher education and socioeconomic status, might have more reluctance to be vaccinated as they have a lower perception of risk for their children. Decision-making for vaccination during pregnancy was variant, fluctuating, and influenced by several factors, the most important being HCWs’ opinions. Besides the clinical protocols, HCWs need personal security and trust to transmit their patient recommendations. Vaccine trust is built on scientific evidence for HCWs, especially midwives, to recommend a vaccine and make clear recommendations with stable protocols that last long enough, as was described before [35]. This last factor was one of the main problems detected during the COVID-19 pandemic, as changes happened fast, and HCWs did not implement those due to mistrust, leading to vaccine hesitancy, especially in certain trimesters of pregnancy. In line with others, midwives were among the primary sources of professional advice for pregnant women [14]. Addressing their understanding and professional practices regarding maternal vaccination is key to changing the attitudes and thus increasing vaccine uptake [14]. It is highly relevant that midwives feel safe and trusted in recommending vaccines. Midwives need more information, ongoing training, and evidence that allows them to recommend maternal vaccination straightforwardly and confidently, particularly in the case of influenza. They also require training in providing information and making vaccine recommendations more effectively. Academic and health institutions should guarantee such training during their basic education and throughout their career to change their conceptions, attitudes, and practices [14].

Mass media influenced social opinions about pandemic management and healthcare systems, as well as perceptions about the need for vaccines. In general, mass media provides positive information about vaccines. However, some participants warned about the need to differentiate trustful sources of information from others to prevent the dissemination of fake news. The media was also perceived as a source of discussions with no expert opinions. Many believed that the role of the media was providing high-quality, evidence-based information and that sensitising the population to health education by experts in the field is needed. Others have concluded that to stop misinformation from eroding public trust in vaccines; there is work needed with social media companies to promote trustworthy vaccine information, provide accurate, accessible information on vaccines to state policymakers, and engage state and local health officials to advance effective local responses to misinformation [26]. Some healthcare workers interviewed advocated for the need to include more holistic health education in society, schools, and media. Regarding clinical trials, uncertainties related to vaccine effects in foetuses or women were one of the main factors for not participating. Perceived risk–benefit is balanced towards more risks than potential benefits. However, they stated the importance of clinical trials to generate evidence before they were vaccinated. Most interviewed women related that the only circumstance when they would participate in a vaccine-clinical trial would be if their foetus presented an anomaly that the trial could cure.

The role of healthcare and medical institutions was key for populations to make their decision-making regarding vaccination. According to Erchick DJ et al., maternity teams should develop protocols to foster social support and patient-centred education around infection prevention that focuses on improved risk perception, expected changes in care due to COVID-19, and vaccine effectiveness and safety [36]. In line with our results, an online cross-sectional survey in the UK concluded that pregnant women preferred receiving vaccine information from their midwives and general practitioners and face-to-face information above other types, such as brochures [37]. It is crucial to provide all types of HCWs in all specialities with updated information about the advantages of vaccines so they can motivate women to feel safe to be vaccinated [38]. During the COVID-19 pandemic, healthcare management has been influenced by political decisions; thus, decisions to be vaccinated have been influenced by personal perceptions about the political system. Thus, building political and social trust is required to restore trust in healthcare systems.

Our study had some limitations. Due to the inherent resources available, we only included women who could speak Spanish, Catalan, French, or Portuguese. Though we believe that there was a wide range of options for the interview to be conducted, we are aware of the potential language barrier for participants who did not understand those languages. The fact that there was only one interviewer and transcriber might also be a limitation. Desirability bias might have existed if the interviewee adapted their narrative to what might be expected by the interviewer, especially for HCWs interviewed, for fear of being recognised (as the sample size was relatively small). However, as the calls advanced, relationships were built, and more rich comments arose, reducing the potential lack of transparency at the beginning. Some recall bias may have also affected some participants’ responses. Participants’ selection bias, however, was minimised by enrolling eligible participants consecutively as they presented for antenatal care visits. All interviews were conducted by phone, which might have been a limitation, as interviewees might have felt less engaged or attached to the interviewer, but it could also be perceived as a benefit, as they might have felt more anonymity and therefore answered more freely. Our results might be interpreted cautiously, as generalisations cannot be performed.

## 5. Conclusions

Pertussis was the most trusted vaccine for pregnant women due to its history of being administered in gestation. The influenza vaccine was conceived as less important and as a non-severe disease. The COVID-19 vaccine was the least trusted by pregnant women due to uncertainties in the mid- and long-term, its fast development, the possibility to avoid infection, the new and innovative technology used, and perceptions of low security due to personal experiences of spontaneous miscarriages attributed to being vaccinated with COVID-19. Despite this, the necessity to be vaccinated has been justified by pregnant women due to the unexceptional situation (COVID-19 pandemic), where protocols have been adapted with the evidence that was immediately generated.

The relationship between pregnant women and HCWs, especially midwives who attend to women during antenatal care visits, is crucial, and plays a key role in vaccine acceptance and uptake. To facilitate this positive relationship, it is essential that the HCWs feel that the vaccination guidelines they receive are trustworthy. Additionally, education through mass media was highlighted as a key component to providing health recommendations and increasing health literacy among the general population.

## Figures and Tables

**Figure 1 vaccines-10-02015-f001:**
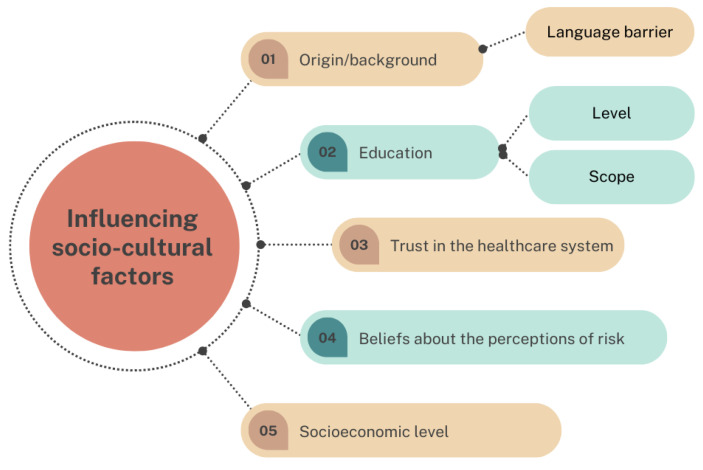
Socio-cultural factors influencing vaccination uptake during pregnancy.

**Figure 2 vaccines-10-02015-f002:**
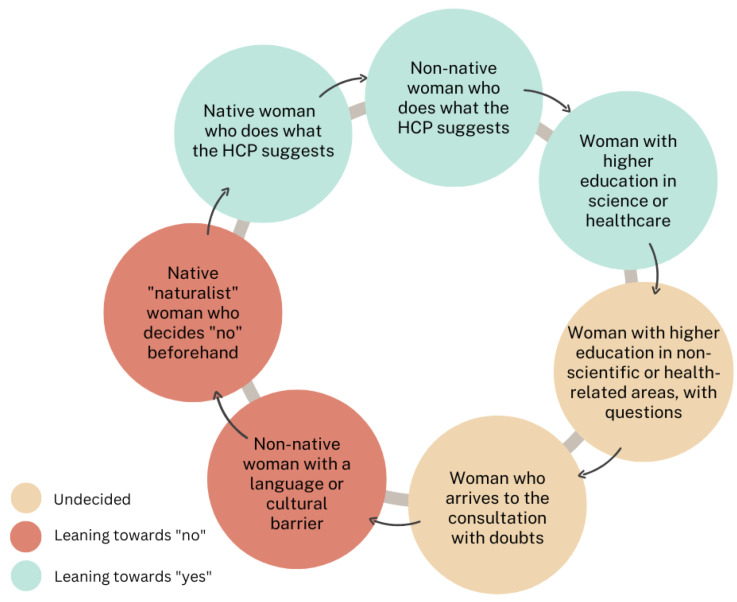
Profiles of pregnant women concerning maternal vaccination.

**Table 1 vaccines-10-02015-t001:** Sociodemographic characteristics of pregnant women and HCWs interviewed.

	Pregnant Women N = 21		Healthcare Workers N = 14	
Characteristics	*n*	(%)	*n*	(%)
**Age (years) ***				
≤25	3	(14.3)	1	(7.1)
26–30	5	(23.8)	4	(28.6)
31–35	4	(19)	2	(14.3)
36–40	6	(28.6)	0	(0)
41–45	3	(14.3)	0	(0)
≥46	0	(0)	6	(42.9)
**Nationality**				
Spain	13	(61.9)	12	(85.7)
Bolivia	1	(4.8)	1	(7.1)
Colombia	2	(9.5)	0	(0)
Honduras	1	(4.8)	0	(0)
Italy	1	(4.8)	0	(0)
Poland	1	(4.8)	0	(0)
The Dominican Republic	1	(4.8)	0	(0)
Venezuela	1	(4.8)	1	(7.1)
**Education**				
Primary	2	(9.5)	0	(0)
Secondary	2	(9.5)	0	(0)
Vocational training	4	(19)	0	(0)
University	13	(61.9)	14	(100)
**Occupation**				
Employed	19	(90.5)	14	(100)
Unemployed	2	(9.5)	0	(0)
**Gestational age (weeks)**				
<24	8	(38.1)	N/A	N/A
24 or more	13	(61.9)	N/A	N/A
**Marital status**				
Single	2	(9.5)	N/A	N/A
In a relationship	9	(42.9)	N/A	N/A
Domestic partnership	3	(14.3)	N/A	N/A
Married	6	(28.6)	N/A	N/A
Widowed	1	(4.8)	N/A	N/A
**Gravidity**				
Primigravidae	6	(28.6)	N/A	N/A
Multigravidae	15	(71.4)	N/A	N/A
**Religion**				
Christian	7	(33.3)	N/A	N/A
None	14	(66.7)	N/A	N/A
**COVID-19 status**				
Had COVID-19 before pregnancy	10	(47.6)	N/A	N/A
Had COVID-19 during pregnancy	10	(47.6)	N/A	N/A
**Medical speciality**				
Gynaecologist/ Obstetrician	N/A	N/A	5	(35.7)
Midwife	N/A	N/A	7	(50)
General practitioner	N/A	N/A	1	(7.1)
Paediatrician	N/A	N/A	1	(7.1)

* One value for age is missing in the healthcare professional category. N/A (Not applicable): These data only apply to one sub-set of interviewees (pregnant women), not to healthcare workers.

**Table 2 vaccines-10-02015-t002:** Motivational factors for deciding to be vaccinated during pregnancy.

Reasons for Accepting Vaccination in Pregnancy	Translated Quotes in English
Good for the baby and the mother.	“The gynaecologist told me about the vaccine. Ehh, she explained it very well. She told me what benefits it would have, what it was good for, ehh that it would help my baby, that it would generate more antibodies in my body. I mean, I didn’t have to think about it much, neither yes nor no. I mean, I really agreed with the vaccination”. 213-01
Avoiding infection.	“I understand that, yes, that all of them can have a risk, but I understand that the risk is low compared to the risk that the disease itself can bring”. 204-01
Good for the community.	“In the first place, the COVID-19 vaccine, because I think we are in a serious pandemic situation and that the need is imminent, right? to stop it (…) So, I think that if, for example, you can get vaccinated and avoid it, then it is important. And if you don’t prevent it with the vaccine, at least perhaps you reduce the effects of, ehh, the virus so that it’s not so severe”. 203-01
Recommendation by healthcare workers.	“Oh, it’s just that no, I mean, for me, it’s not… I mean, I haven’t considered a rejection, I mean, not getting vaccinated for me, it was a matter of definitely getting vaccinated. I mean, it wasn’t… it’s not debatable. (…) if a doctor tells me that I have to get this, well, I think he is the one who knows the criteria so that, so that I take a medicine or a vaccine or whatever”. 214-01
Trust in the healthcare system.	“…I think that it [maternal vaccination] greatly depends on your opinion, on your trust in the healthcare system, and the trust in vaccines, medicines, treatments, well, all these things”. 204-01

**Table 3 vaccines-10-02015-t003:** Motivational factors for deciding not to be vaccinated during pregnancy.

Reasons for Refusing Vaccination in Pregnancy	Translated Quotes in English
Fear that it could affect foetal development.	“There are people who are very distrusting and very, very suspicious. But it is true that pregnant women are perhaps obviously afraid that vaccines can affect the development of the foetus and that they don’t see these advantages, do they? To protect themselves from all the infections that others may have that can pass to the foetus through the placenta”. 207-02
Low perception of disease risk.	“Because I thought I wouldn’t… no, no… get vaccinated… I didn’t have to save myself from getting the flu. And if I got the flu, well, well, I’d be in bed for a week with a stuffy nose. But hey… nothing happens, right? (…) I did not see the need, the usefulness”. 203-01
Distrust in the healthcare system.	“Their arguments are very irrational, “what if they think we are here to be guinea pigs”, “what if not… we don’t have any kind of trust”. I mean, in fact, I anything, “I knew a person who was vaccinated and look what happened to them…”. (…) They don’t listen and they’re making generalisations or having a very catastrophic outlook or considering a very suspicious view”. 207-02
Not being able to counteract the side effects with medication.	“Man, I think… well, let’s see, what can affect (the decision) a little more is that if you feel sick, so to speak, or you get a fever, maybe because you can’t have as many medications or don’t have them… yes you can take paracetamol. But hey, it’s not very appropriate for you to take it either. Well, maybe going through those symptoms without anything is what can cause doubts” 205-01
Political views/Moral values.	“They say that the vaccines are not safe or that it is manipulation by the pharmaceutical companies. (…) And at the end of the day, they also don’t see individual benefits, don’t they? Because if it has to be a group benefit, then they don’t see it” 205-02
Not believing in vaccines or other medications.	“Generally, people who do not want to be vaccinated do not want other medications either, that is, if they have diabetes, they would not inject insulin either. In other words, it is usually people who are already reluctant to any medication for fear that…” 105-02
Fear of contracting an illness through vaccination.	“[Somebody] got the flu vaccine one day and they think they developed flu, which is not true because they cannot develop it because it is an inactivated vaccine. But… but well, it’s their belief. Then these people are more reluctant to get the vaccine”. 102-02

## Data Availability

The data presented in this study are available on request from the corresponding author. The data are not publicly available due to confidentiality issues with interviewees.

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
