# Peer review of "“Maternal Vaccination Greatly Depends on Your Trust in the Healthcare System”: A Qualitative Study on the Acceptability of Maternal Vaccines among Pregnant Women and Healthcare Workers in Barcelona, Spain"

_vaccines, 2022, doi:10.3390/vaccines10122015_

Round 1
Reviewer 1 Report
This study illustrated a scenario of vaccination among pregnant women and HCPs in Barcelona, Spain. The findings may facilitate the improvement of vaccination for pregnant women, which remains a challenge in public health and clinical practice.
1. The interview included limted pregnant women, among which 1/3 had foreign nationalities other than Spain. I wonder if the diversity might result in different perception and awareness of vaccination. Please add more explanation and discussion.
2. I suggest the authors may present more description of perception among HCPs, which seems less than that among pregnant women in the manuscript. Also, the authors may compare the perceptions between pregnant women and HCPs, and further identify potential consistency or inconsistency.
3. I wonder if the authors select only pertussis, influenza, and COVID-19 vaccines as questions in the interview, as adults including pregnant women may receive more vaccines. Please add an interview outline in the supplementary materials.
Author Response
Thank you very much for sending such constructive and positive feedback.
Please, find here our responses point by point:
- Sample:
1.1 We understand that for an epidemiological scientific article, the sample size might be very small. However, we would like to remind the reviewer that this is a scientific article performed with a qualitative methodological approach. The sample size was defined on the concept of saturation point. Thank you for the reviewers’ suggestion, and following the article “Bowen, 2008, Naturalistic inquiry and the saturation concept: a research note”; we have added an explanation in the manuscript to explain how the saturation point was reached. Now we have included the following paragraph: “The sample size was defined based on a saturation point whereby all themes had been thoroughly explored, and no new themes emerged in subsequent interviews. During the coding process, and comparing categories that emerged during the interviews, as all themes emerged in both groups, the saturation point was achieved; and no more interviews were scheduled to be performed.”
1.2 Regarding the question of the diversity, women interviewed were randomly selected. The rate of 1/3 of women being of foreign nationalities represents the multiculturality of Barcelona, where the study was performed. It might not be representative of the whole country, as each region might have variations in their populations. However, the trends that we saw in our community have been detailed in the sub-section 3.3.2.
- Thank you for this feedback. We have improved the Results and Discussion sections to include more information about healthcare workers perceptions, and the comparison with pregnant women’s experiences.
- We have included the maternal vaccines that are recommended and administered at antenatal care visits in Spain, which at the moment are flu, Tdap, and COVID-19. We also explored in the interviews pregnant women’s acceptability to participate in a clinical trial including other maternal vaccines (as described in the results, hesitancy to participate in them was very high). We have included the interview guide as supplementary materials.
We would like to thank the reviewer for these suggestions that helped us improve the quality of the manuscript. We hope that the changes made in the manuscript meet reviewer expectations.
Sincerely,
Reviewer 2 Report
Dear Authors,
I have read and mostly appreciated the manuscript titled “Maternal vaccination depends a lot on your trust in the healthcare system”. Acceptability of maternal vaccines among pregnant women and healthcare professionals in Barcelona, Spain: A qualitative study, which is based on semi-structured interviews of pregnant women attending antenatal care services and healthcare professionals in Barcelona, Spain.
Firstly, I have to point out that the title comes across as muddled and needs to be rephrased. Starting off with a quote is not advisable when utting together a scientific paper.
The article is praiseworthy in terms of relevance, data analysis and structuring; it relies on sound methodology which has been well expounded upon by the authors. More info on the methodology, however, should be included in the abstract. Data analysis is convincing and the tables/figures are effective at conveying key findings.
The interspersed, scattered quotes at 3.2 are not acceptable, because they seem to be arbitrarily placed following what seems to be part of a discussion, then other quotes come within table 3. That has to be rationalized.
Furthermore, I would strongly recommend adding further elements of discussion on the highly complex and multifaceted issue of vaccine hesitancy.
There is no mention of vaccine mandates when the relevance of trust in healthcare authorities is discussed, although they are likely to have a bearing on attitudes towards the vaccines themselves, or of how the COVID-19 pandemic may have changed attitudes towards vaccinations in pregnancy.
Please consider drawing upon and citing the following sources:
doi: 10.26355/eurrev_202201_27891.
doi: 10.1080/14767058.2022.2128652
doi: 10.1111/1471-0528.17110.
doi: 10.3390/vaccines9101107
doi: 10.1016/j.ajog.2021.08.007
All in all, I believe the article has strengths and is competently assembled, despite a few inconsistencies which need to be addressed, as pointed out earlier.
Lastly, although well written overall, the article needs further proofreading by a native speaker of English. Some sentences are somewhat convoluted and need to be rephrased for the sake of clarity and readability.
Sincerely.
Author Response
Thank you very much for sending such constructive and positive feedback.
Please, find here our responses point by point:
- As this is a qualitative scientific article, we have included a quote in the title to catch the attention of the readers. Though this is not done for quantitative scientific papers, it is a common practice in scientific articles using a qualitative methodology. For that reason, we would prefer to leave the quote in the title, if the reviewers agree.
- Regarding the methodology, we have added a sentence to detail better the analysis in the Abstract, now it reads: “This qualitative study was conducted based on phenomenology, a methodological approach to understand first-hand experiences, and grounded theory, an inductive approach to analyse data, where theoretical generalisations emerge. Data were collected through semi-structured interviews with pregnant women attending antenatal care services and healthcare professionals in Barcelona, Spain. Interviews were audio-recorded, transcribed, and coded; and notes were taken. Inductive thematic analysis was performed, and data were manually coded.”
- The quotes under section 3.2 illustrate the messages from that section in a qualitative manner. We used quotes to illustrate the thematic analysis. As a way to include quotes in different ways, not only in Tables or Figures, we included them within the Results section. This is a common practice in qualitative articles, and we would appreciate leaving them there. However, if the reviewer disagrees, we can delete the quotes from the text.
- We agree with the reviewer that this is a very complex and multifaceted issue. We have improved the discussion section with the articles provided, including a discussion about the vaccine mandates. Thank you very much for the references.
- Lastly, just to clarify, that one of the main authors is a native English speaker and certified copy editor and proof-reader with more than five years of experience reviewing scientific articles.
We would like to thank the reviewer for these suggestions that helped us improve the quality of the manuscript. We hope that the changes made in the manuscript meet reviewer expectations.
Sincerely,
Reviewer 3 Report
Strength of study,
The authors provided a detailed literature review and a well-designed study on maternal immunization concerns. The study adds valuable data to the literature, especially by revealing the hesitancies of the Spanish-origin local population about vaccination during pregnancy.
Limitation of study,
The number of subjects participating in the study is too small. This is a major limitation. Language problems of the participants may also have negatively affected the study.
1. Too few subjects were included in the study. Were statistical evaluations of power and sample size made before the study?
2. Was there a difference between trimesters in terms of vaccine hesitancy?
Author Response
Thank you very much for sending such constructive and positive feedback,
Please, find here our responses point by point:
- We undestand that language barriers could exist, as something inherent in qualitative interviewing. However, we are confident that the language does not seem to be a limitation for us, as the interviewer was fluent in 4 different languages (including the 2 official languages of the region where the study was conducted), allowing the majority of individuals to participate in them. As in all studies performed in non-English speaking countries, after the interviews take place, a process of translation to English occurs to share the findings with the scientific community. One of the main authors of this manuscript is a native English speaker, and certified-proof reader of scientific articles with more than five years of experience.
- Statistical evaluations are not performed in qualitative studies. We would like to highlight to the reviewer that this study was not an epidemiological, or other type of quantitative survey-based study, but a qualitative one based on in-depth experiences and perceptions from few participants. The sample size was defined was on the concept of saturation point. Thank you to the reviewers’ suggestion, and following the article “Bowen, 2008, Naturalistic inquiry and the saturation concept: a research note”; we have added an explanation in the manuscript to explain how the saturation point was reached. Now we have included the following paragraph: “The sample size was defined based on a saturation point whereby all themes had been thoroughly explored, and no new themes emerged in subsequent interviews. During the coding process, and comparing categories that emerged during the interviews, as all themes emerged in both groups, the saturation point was achieved; and no more interviews were scheduled to be performed.” The following references have been added to support this concept of the saturation point:
- Thyer, B. The Handbook of Social Work Research Methods, 2nd ed.; SAGE Publ Inc.: Thousand Oaks, CA, USA, 2010.
- Glaser, B.G.; Strauss, A.L. The Discovery of Grounded Theory: Strategies for Qualitative Research; Sociology Press: Mill Valley, CA, USA, 1967.
- Bowen, G.A. Naturalistic inquiry and the saturation concept: A research note. Res.2008, 8, 137–152.
- Differences in trimesters of pregnancy:
- This study was not performed with the aim to statistically compare sociodemographic factors and participants responses. As previously highlighted, this is a qualitative study performed with the aim to explore and further understand participants’ experiences and perceptions about vaccinations in pregnancy. After highlighting that, we can discuss that indeed vaccination hesitancy depended on the trimester of pregnancy, firstly because pregnant women receive the information about the vaccines depending on the trimester of pregnancy when vaccines are recommended (i.e., HCW do not provide all the information at the beginning of pregnancy, but little by little as the pregnancy evolves, and the potential window for vaccination is closer). Secondly, because perceptions of risks of potential harm to the foetus are not the same in all trimesters (i.e., as the foetus evolves, there is less risk of neurological complications).
- As we cannot make correlations, we have included an article that performed an statistical evaluation in the discussion section. (page 13, likes 460-462): “In Texas (USA), a cross-sectional survey study on the sociodemographic predictors for COVID-19 vaccine hesitancy in pregnancy found that women who were hesitant were younger, had a more advanced pregnancy, and were also hesitancy for flu and Tdap vaccination.”
We would like to thank the reviewer for these suggestions that helped us improve the quality of the manuscript. We hope that the changes made in the manuscript meet reviewer expectations.
Sincerely,
Round 2
Reviewer 1 Report
The manuscript is well revised for publication.
Reviewer 2 Report
Dear Authors,
I can certainly appreciate the extent to which you have managed to improve your article.
The additions are valuable in terms of adding elaboration and contextualization for shedding a light on the numerous contributing factors and complexities which shape and determine vaccination hesitancy and acceptance.
Please reduce the length of the very long quotes at 3.2, leaving only the most representative/essential remarks.
I recommend further proofreading for the sake of clarity and readability, although the article is fairly well-written overall.
Lastly, you need to make the references compliant with MDPI style.
Sincerely.